# Modelling the Human Placental Interface In Vitro—A Review

**DOI:** 10.3390/mi12080884

**Published:** 2021-07-27

**Authors:** Marta Cherubini, Scott Erickson, Kristina Haase

**Affiliations:** European Molecular Biology Laboratory (EMBL), 08003 Barcelona, Spain; marta.cherubini@embl.es (M.C.); scott.erickson@embl.es (S.E.)

**Keywords:** placenta, maternal-fetal interface, trophoblast invasion, bioprinting, microfluidics, placenta-on-a-chip, in vitro models

## Abstract

Acting as the primary link between mother and fetus, the placenta is involved in regulating nutrient, oxygen, and waste exchange; thus, healthy placental development is crucial for a successful pregnancy. In line with the increasing demands of the fetus, the placenta evolves throughout pregnancy, making it a particularly difficult organ to study. Research into placental development and dysfunction poses a unique scientific challenge due to ethical constraints and the differences in morphology and function that exist between species. Recently, there have been increased efforts towards generating in vitro models of the human placenta. Advancements in the differentiation of human induced pluripotent stem cells (hiPSCs), microfluidics, and bioprinting have each contributed to the development of new models, which can be designed to closely match physiological in vivo conditions. By including relevant placental cell types and control over the microenvironment, these new in vitro models promise to reveal clues to the pathogenesis of placental dysfunction and facilitate drug testing across the maternal-fetal interface. In this minireview, we aim to highlight current in vitro placental models and their applications in the study of disease and discuss future avenues for these in vitro models.

## 1. Introduction

The human placenta is a crucial organ that supports fetal development throughout gestation. Placental growth and function are precisely regulated to ensure effective circulation of oxygen and nutrients, removal of waste, generation and release of metabolites, and protection against diseases, infections, and xenobiotic transfer to the fetus [1]. Considering its vital role, it is essential to understand placental development and the causes of its dysfunction. However, due to ethical concerns, our understanding of the placenta is largely derived from explants at term or from unsuccessful pregnancies. Explants have provided many clues into pathological pregnancies, such as fetal growth restriction, pre-eclampsia, and stillbirth at varied stages of disease [2,3,4]. However, explants begin to degenerate within hours after collection, making experimentation with human tissue challenging. Efforts have been made to develop accurate animal models [5]; however, considerable differences between species make it difficult to develop a non-primate animal model that fully mimics human placentation [6,7]. Rodent models are useful for understanding specific aspects of placentation, but many processes are difficult to assess in vivo. Ultimately, bioengineered in vitro models promise to bridge the gap between species and offer precise control over the microenvironment to recapitulate specific aspects of human placentation in health and disease [8].

This minireview aims to: (1) discuss our current understanding of human placentation and highlight areas that require further investigation; (2) discuss current in vitro placental systems, ranging from 2D to 3D models; (3) explore recent applications of these models in studying placental physiology and disorders; and (4) discuss key components to consider when developing or evaluating an in vitro model of the placenta.

## 2. Development and Functions of the Human Placenta

The placenta, a fetal organ, forms shortly after fertilization and continues to change throughout pregnancy in response to the metabolic demands of the fetus. Placentation begins post-fertilization when the blastocyst attaches to the inner layer (endometrium) of the uterus. The blastocyst then begins to invade the endometrium with the help of its outer layer of cells, termed trophoblasts. This fetal trophoblastic layer is divided into two cell types, an external multinucleated syncytiotrophoblast layer (the invasive trophoblasts) and an inner cytotrophoblast layer. About two weeks after fertilization, the external syncytiotrophoblast forms preliminary fluid-filled villi structures directed outward, towards the decidual layer of the mother’s uterus (Figure 1a). Then, the cytotrophoblasts proliferate and migrate through the syncytiotrophoblastic layer to form the primary villi [9]. Soon, these villi expand and become vascularized with fetal placental vessels. Meanwhile, trophoblasts remodel the maternal spiral arteries of the decidua, which become dilated, allowing for maternal blood to fill the intervillous space. As a result, there is a large surface area for the exchange of nutrients traveling from the mother’s circulation into the intervillous spaces, through the trophoblast layers, and into the closed placental circulation of the villi, which nourishes the fetus via the umbilical cord [10,11]. By the second trimester, the main features of the mature placenta are formed (Figure 1b).

Nutrient exchange between maternal and fetal blood is facilitated largely by the syncytiotrophoblast, which is one continuous multinucleated layer of cells (Figure 1c). The ability of nutrients to cross this layer depends on transporter proteins and its thickness, which is reduced near the vascularized parts of the villi [12]. Small hydrophobic molecules, such as oxygen and carbon dioxide, can easily diffuse across plasma membranes in response to differences in the concentration gradient between maternal and fetal blood, which varies with the maternal blood supply, environment, and rate of blood flow. Partial pressure of oxygen in the maternal blood is considerably higher than in the fetal blood, while carbon dioxide is more abundant in the fetal blood. Oxygen exchange is also facilitated by fetal hemoglobin having a higher affinity for oxygen than that of an adult [13]. Consequently, oxygen diffuses through the placenta from the maternal to the fetal blood, while carbon dioxide diffuses in the opposite direction. Transport of large (molecular weight > 1 kDa) and hydrophilic molecules, however, is size restricted and diffusion limited, therefore, depending on the presence of various transporter proteins to increase flux [14,15].

The placenta also functions as an immunological barrier, countering the maternal immune response that would normally cause rejection of the fetus, finally leading to spontaneous abortion [16]. Still, maternal antibodies (immunoglobulin G (IgG)) are actively transported across the placenta by neonatal Fc receptors (FcRns), conferring protection against infections to the fetus and the neonate during the first months of life [17].

In addition to its barrier function, the placenta acts as an endocrine organ. For example, to prevent the progression of the menstrual cycle and the loss of the endometrial lining, the syncytiotrophoblast releases human chorionic gonadotropin (HCG). HCG prolongs the life of the corpus luteum, which is thus able to continue releasing progesterone and promote the healthy function of endometrial vasculature, preventing its deterioration and loss [18]. The placenta also produces placental growth hormone (PGH), which is structurally very similar to pituitary growth hormone and eventually completely replaces it [19]. Importantly, the placenta supports pregnancy and fetal growth by selectively secreting the steroids estrogen and progesterone [19].

Although much of placental physiology has already been characterized, there are many aspects that are not well understood. For instance, little is known about the cellular mechanisms that drive the programmed events of placental branching angiogenesis, the regulation of the permeability of the maternal-fetal interface, the behavior of extravillous trophoblasts, and the impact of nutrients, hormones, and environmental factors on placental development. In the next section, we discuss the latest in vitro systems that aim to recapitulate important features of the human placenta and eventually provide answers to these questions, with a particular focus on mimicking the maternal-fetal interface and trophoblast invasion.

## 3. In Vitro Models of the Placental Barrier

A variety of in vitro models have been developed to study different aspects of placental biology; however, one aspect of particular focus is the maternal-fetal interface as a barrier (Figure 2a). Cells derived from gestational choriocarcinoma [20,21] or immortalized trophoblasts [22] have been widely used to represent both villous and extravillous trophoblasts as cultured monolayers. The monolayers have been grown on plates or semipermeable membranes (transwell inserts) and have been employed to investigate hormone secretion, transcellular transport of glucose, environmental toxicants, and susceptibility to parasite infection [23,24,25]. Despite their simplicity, cell monolayer models can be an effective first step to studying the mechanisms and properties of the human placental barrier. However, by focusing on just one aspect of the placenta (trophoblasts), these models lack physiological complexity and a comparable cellular microenvironment. Recently, Kreuder et al. addressed some of these drawbacks by including essential components of the placental villi, such as fibroblasts and endothelial cells [26], as represented in Figure 2b. Their model involved bioprinting a methacrylated gelatin membrane (GelMA), which mimics extracellular matrix (ECM) features, containing primary placental fibroblasts, to simulate villous stroma. BeWo trophoblasts and primary human placental endothelial cells were cultured on either side of this printed membrane, thus representing a more complex model of human placental villi [26]. Barrier properties were assessed by two permeability assays: one using a fluorescently labeled molecule to measure solute flux, and the other by impedance-based measurements using a transepithelial electrical resistance (TEER) system. Their results showed that the bioprinted membrane presents physiological ECM-like features, such as a lower elasticity, which resembles that of placental tissue, in comparison with filter membrane-based systems. Moreover, TEER values were higher when BeWo trophoblasts were cultured on the membrane containing fibroblasts rather than in monotypic cultures, demonstrating a reduction in permeability (reduced leakiness) due to the incorporation of the stromal compartment.

In vitro models of the placental barrier have also recently incorporated 3D vascularized networks. For example, Nishiguchi et al. used a modified transwell system to seed a layer of laminin and collagen-coated trophoblasts (either BeWo or primary cytotrophoblasts) onto a thick layer of self-assembled capillary networks, formed from primary fibroblasts (normal human dermal fibroblasts (NHDFs)) and human umbilical vein endothelial cells (HUVECs) in a fibrin hydrogel [27] (Figure 2c). This model was employed to examine cell damage signaling across the barrier by exposing rat embryonic cortical neurons to conditioned medium collected from the in vitro placental barriers assembled with direct or indirect contact with the vascular bed. Although the vessels were not perfused, the presence of the vasculature resulted in a reduction in neuron dendrite length, providing evidence of crosstalk between the trophoblasts and the endothelium.

Considering the multilayered structure and physiologic microenvironment critical to placental barrier function, other groups have also attempted to generate more complex models, including both an endothelial (representative of the fetal vessels) and trophoblast component. For instance, cocultures of endothelial and trophoblast cells have been combined in sandwiched monolayers on chip, which allow for exposure to fluid flow mimicking the hemodynamic shear stress present in maternal and fetal compartments (Figure 2d). Blundell et al. generated a two-layer polydimethylsiloxane (PDMS) device with two channels separated by a thin porous membrane, which allows for constant perfusion with culture media [28]. BeWo trophoblasts were cultured on the upper side of the membrane and human placental vascular endothelial cells (HPVECs) on the lower side. This coculture model recapitulated structural features of the maternal-fetal interface and showed the expression and physiological localization of placental transporter proteins. The authors observed more complete formation of dense microvilli projections on the apical surface of the trophoblast cells when cultured under fluid shear stress conditions, when compared with static culture. Moreover, they found that inclusion of the fetal endothelium was crucial to replicate physiological maternal-fetal glucose transport, as confirmed by comparing with the glucose transfer rates measured across two other types of barriers: a cell-free barrier and a trophoblast monolayer without endothelium [28]. In a similar approach, Lee et al. cultured JEG-3 trophoblasts and HUVECs on either side of a solidified collagen membrane and subjected each side to dynamic flow conditions. The system facilitated cell proliferation and the formation of confluent monolayers into a placental barrier model, which demonstrated different glucose transport rates depending on the presence of the epithelium and in accordance with findings from Blundell’s model [17]. More recently, a similar microfluidic two-channel design with a polyethylene (PETE) membrane separating monolayers of BeWo trophoblasts and HUVECs was tested to examine caffeine transport across the placenta, a molecule that cannot be fully metabolized by a developing fetus [29]. This study provided new insights into the extent of caffeine transfer from mother to fetus and demonstrated the utility of the system for future xenobiotic compound testing.

Another way to achieve the complex geometry of the placental villous membrane, while bypassing the use of flat cell monolayers, was proposed by Mandt et al., who developed a barrier model using a high-resolution three-dimensional (3D) printing method called two-photon polymerization (2PP) [30]. A villi-like convoluted surface within a microfluidic device with two separate channels was shaped by 2PP from a modified gelatin-based hydrogel material (GelMA), mimicking the basal membrane of the placenta (Figure 2e). To mimic the fetal and maternal compartments, HUVEC and BeWo trophoblasts were then seeded on either side of the membrane and cultured under constant flow. The authors studied transcellular transport across this barrier and demonstrated in vivo-like properties by showing the permeability of sugar-sized molecules (riboflavin, 350 Da) and the impermeability of larger ones (dextran, 200 kDa).

## 4. In Vitro Models of Trophoblast Invasion

Improper trophoblast invasion into the endometrial spiral arteries is often associated with pregnancy complications, including pre-eclampsia and fetal growth restriction [31]. Thus, understanding how spiral artery remodeling impacts the early steps of placental development is crucial. A variety of in vitro models specifically designed to replicate the process of trophoblast invasion (Figure 3a) have been developed. Transwell assays have been extensively used to assess trophoblast invasiveness and generally involve the observation of cell migration through a Matrigel layer, often towards a chemoattractant (Figure 3b) [32,33,34,35]. However, these 2D adherent cell systems do not fully replicate the invasion process in an anatomically relevant manner. Recently, the inclusion of self-assembled spheroids of extravillous trophoblasts has allowed for complex 3D cell-cell interactions, bringing important insight into the mechanisms underlying trophoblast migration and invasion (Figure 3c). For instance, You et al. demonstrated that endometrial signaling is essential to promote and guide trophoblast invasion by observing that trophoblasts were able to migrate from the spheroids and invade the Matrigel only when there was an underlying layer of human endometrial stromal cells [36].

Bioprinting allows for controlled 3D spatial patterning of cells, biomaterials, ECM components, and growth factors in order to generate tissue analogues and has led to the development of a number of physiologically relevant trophoblast invasion assays (Figure 3d). One such study utilized bioprinting to generate a cylindrical hydrogel model containing placental basement membrane (BM) proteins, including collagen, laminin, and fibronectin, and a central source of chemoattractant (epidermal growth factor (EGF)) at the center and an outer layer of cytotrophoblasts [37]. The results showed that trophoblast cell migration was significantly higher in the presence of BM proteins when compared with the empty hydrogel, demonstrating the importance of the ECM microenvironment in trophoblast invasion. Similarly, Ding et al. bioprinted multiring and multistrip hydrogel systems incorporating EGF and adjacent layers with and without encapsulated cells (invasive trophoblasts, HTR-8/SVneo). Their strategy enabled the recapitulation and modulation of in vivo 3D cellular microenvironments and the study of trophoblast migration in different geometries [38]. Considering that EGF is downregulated in pre-eclampsia (PE) [42], Kuo et al. generated a cylindrical 3D-printed GelMA hydrogel loaded with different concentrations of the growth factor to study the migratory response of trophoblasts in the development of PE. Their results showed that trophoblast migration increases in response to higher EGF concentrations [39]. Since insufficient trophoblast invasion is a primary feature of PE, this model represents a useful tool in the identification of novel therapeutic targets for its treatment.

Besides bioprinting, microfluidic models have also been useful in tracking and quantifying the dynamics of trophoblast cell migration since they can be designed to generate stable gradients on chip (Figure 3e). As an example, Abbas et al. embedded primary trophoblasts in Matrigel and demonstrated the impact on their migratory behavior by including a gradient of granulocyte-macrophage colony-stimulating factor (GM-CSF) [40]. In fact, with the addition of the gradient, trophoblast cells exhibited increased directionality and motility, suggesting that GM-CSF is a key cytokine in the regulation of trophoblast invasion. Recently, the system was improved by including endothelial cells to elucidate their effect on trophoblast invasiveness [41]. Moreover, invasion-stimulation was induced with folic acid, and trophoblast tracking was facilitated by the incorporation of fluorescent cell tagging. The results showed that trophoblast invasiveness was enhanced in the presence of endothelial cells, suggesting that the release of cytokines and growth factors from the endothelium has a role in trophoblast migration.

## 5. Three-Dimensional Models to Study Placental Dysfunction, Infections, and Maternal-Fetal Toxicology

Besides the defective remodeling of the spiral arteries, many other aspects of placental dysfunction are associated with altered placental development and the onset of pregnancy complications. These events, which include the impairment of villous tree maturation [43], the detrimental effects of pathogen infections [44], and the response to drugs and environmental cues [45,46], still need further investigation. As discussed previously, the advent of technologies such as bioprinting and microfluidic-based organs-on-a-chip have facilitated the recapitulation of critical placental functions and stages of development, raising the possibility to apply these models to study and elucidate the mechanisms underlying these aberrant events. For example, a previous work from our group brought new insight into placental vasculopathy, showing that pericytes (mural cells of the microvasculature) contribute to growth restriction of fetal microvessels grown in microfluidic devices [47]. Moreover, the results showed PE-like effects, including upregulation of inflammatory cytokines, hyperproliferation of stromal cells, dysfunctional barrier properties, and immune cell infiltration.

Although the placenta acts as a barrier to many pathogens and viruses, rubella virus, cytomegalovirus, herpes simplex virus, Zika virus, and parasites such as *Plasmodium falciparum*, all can cross the placenta and cause adverse birth outcomes [48,49]. To date, the mechanisms leading to infection-driven defects at the maternal-fetal interface are yet to be fully established. Recently, a study conducted in a 3D-based culture model using human JEG-3 trophoblast cells and human microvascular endothelial cells was used to study placental resistance to toxoplasmosis and vesicular stomatitis virus (VSV) infection [50]. Physiological levels of fluid shear stress, produced by a rotating wall vessel (RWV) bioreactor, were able to mimic in vivo syncytiotrophoblast features by inducing spontaneous syncytia fusion events and expression of syncytiotrophoblast markers. Interestingly, the authors found that only the 3D cocultured aggregates exhibited viral and microbial resistance when compared with 2D monotypic cultures.

Microfluidic technologies have also been implemented to explore the impact of pathogenic infections during pregnancy. Zhu et al. generated a multilayered microfluidic placental barrier-on-a-chip model to investigate the placental inflammatory responses to bacterial infection [51]. When *Escherichia coli* was applied to the maternal side of the chip, trophoblast cells triggered an acute inflammatory response by secreting interleukin-1α, IL-1β, and IL-8 cytokines, followed by the adhesion of maternal macrophages. More recently, a microfluidic organ-on-a-chip model comprising the decidua and the fetal chorionic and amnionic membranes was generated to track the propagation of infection and inflammation across the maternal-fetal interface [52]. This four-chamber system containing primary cells from the maternal-fetal interface and a collagen matrix mimics cellular features seen in the native tissue, such as morphology, cellular transitions, migration, and production of nascent collagen. The ascending infection and consequent inflammation were tracked by examining the propagation of lipopolysaccharide (LPS) from the decidua to the amnion. The results demonstrated the disruption of the maternal-fetal interface integrity during ascending infection due to an imbalanced immune response, an event that is associated with preterm birth [53].

It is particularly challenging to study the effects of drugs on the structure and function of the placental barrier since they cannot be tested on pregnant women. The thalidomide disaster of the 1960’s [54] unveiled that the placenta is not an impenetrable barrier and allows xenobiotics to cross from the maternal to the fetal circulation, leading to congenital abnormalities. Therefore, it has become crucial to ensure that potential therapeutic agents and common medications do not impact human fetal development. To overcome this, placenta-on-a-chip technologies have been adopted to study the transport of drugs across the placental barrier, demonstrating their potential use as preclinical drug efficacy and drug safety testing tools. For example, the microfluidic model by Blundell et al., described earlier, was used to investigate the diffusion of heparin and the gestational diabetes drug glyburide across the maternal-fetal interface [55], demonstrating the capability to recapitulate the native function of efflux transporters and the limited drug intake of an in vivo placenta. Finally, increasing evidence also indicates that nanoparticles can cross the placenta barrier, eliciting a toxic effect [56]. For instance, the impact of exposure to titanium dioxide, a common nanomaterial used in plastics, medicines, food products, cosmetics, and toothpastes, has recently been investigated using a micro-engineered 3D placental model [57]. The results showed disruption of placental barrier integrity and adhesion of maternal immune cells in the presence of this nanomaterial.

Overall, the employment of 3D micro-engineered models creates a suitable and controlled approach towards the understanding of the effects of drug treatments and disease conditions on placental function.

## 6. Engineering an Ideal Human Placenta-on-a-Chip

Recent bioengineered placental models have enhanced the knowledge in the field; however, there are a number of areas that should be considered to physiologically recapitulate the human placenta in health and disease (Figure 4). The use of immortalized and carcinoma-derived cell lines (BeWo, JEG-3, and HTR-8/SVneo) for the generation of the trophoblast epithelium has been recurrent in the field due to their relative ease of use. These cells are certainly useful, but they represent specific differentiation pathways of trophoblasts during placentation and so may show alteration of their native functions and responsiveness to stimuli in vitro. As one alternative, primary cells isolated directly from fresh tissue samples can be cultured without losing their morphological and functional features [58]. Numerous procedures to isolate, culture, and characterize cells from various parts of the placenta have been well established [59,60,61,62,63]. Implementation of primary cultures obtained from women with pregnancy disorders will contribute to a better understanding of the mechanisms underlying placental dysfunction. However, primary cells cannot be maintained for long-term experiments since they undergo senescence processes. To address this, human embryonic stem cells (hESCs) and induced pluripotent stem cells (iPSCs) are now being employed to reproduce placental structure and function in vitro due to their unique stemness and their ability to differentiate towards relevant cell phenotypes (both extravillous and syncytiotrophoblasts) [64,65,66]. Importantly, patient-derived iPSCs have represented a useful resource to more accurately model placental disease and improve diagnostics and drug discovery. For instance, some advanced studies have generated trophoblast cells by differentiating iPSCs derived from placentas of normal and pre-eclamptic pregnancies [67,68]. These cells exhibited phenotypic features associated with PE, such as reduced invasive capacity [67], proving to be a valid source for the generation of models of pathological placental development.

Together with the appropriate cell types, an adequate structural and spatial arrangement of the cells (e.g., villi-like structure) is needed to mimic, on-chip, the in vivo maternal-fetal interface in a more physiologically relevant way. Moreover, normal tissue development and homeostasis are critically regulated by biomechanical cues, such as stretching, compression, and matrix stiffness [69]. To date, the role of substrate mechanics in placental development and disease is poorly characterized. Importantly, in vivo studies found that placenta stiffness and elasticity are increased in pregnancy disorders, when compared with normal gestations [70,71]. Thus, the generation of patterned biomaterial matrices including ECM components (collagens, laminin, fibronectin, glycoproteins), with rigidity similar to that of the placental connective tissue, may provide further insight into placental development and dysfunction. One recent study highlighted the relevance of substrate tension, demonstrating that trophoblast fusion and function are affected when cultured on polyacrylamide hydrogels with stiffness resembling pre-eclamptic placental tissue [72]. Another approach being employed to more closely model the structural characteristics of the human placenta are organoids. For example, Turco et al. isolated first-trimester villi trophoblast stem cells to generate 3D organoids that differentiate into both extravillous and syncytiotrophoblasts and form villous-like structures. Furthermore, they also exhibited secretome properties of fetal villi, as demonstrated by the release of pregnancy hormones such as HCG [73].

Despite the important achievements in the recapitulation of structure and cell composition of the maternal-fetal interface, the presence of perfusable vasculature remains a missing feature of current placental models. Vascular hemodynamics not only are essential to transport nutrients, oxygen, and metabolic waste but also play a key role in placental physiology. Indeed, in vivo, blood flow imparts mechanical stimulation on the endothelial wall of the vessels (shear stress), which is essential to the development and establishment of the uteroplacental and fetoplacental circulations [74]. As reported here, the majority of existing models employ a monolayer of endothelial cells cultured in transwells or microfluidic systems to reproduce the wall of the fetal vessels. However, this approach cannot represent angiogenesis and vascular remodeling events that occur during early steps of placental development [75]. This becomes even more relevant when investigating placental dysfunction since the pathological remodeling of vascular tissue underlies the appearance of pregnancy complications. To improve this, several methodologies (patterned microchannels, sacrificial molds, or self-assembled networks) have been developed to generate perfusable vasculature-on-a-chip [76]. Physiological flow rates (interstitial and luminal) can be applied and regulated in these systems via hydrostatic pressure gradients, peristaltic pumps, or syringe pumps.

Another element commonly missing from current placental models is the inclusion of immune cells, in particular fetal villus macrophages (Hofbauer cells), whose function within the stroma has not yet been fully elucidated [77]. Hofbauer cells are some of the most abundant immune cells in the human placenta and are thought to play an important role in angiogenesis and remodeling [78]. Perturbed function and abundance of macrophages are associated with pregnancy complications, as in cases of chorioamnionitis, pre-eclampsia, gestational diabetes mellitus, and severe outcomes associated with viral infection [77]. To date, phenotypic and functional characterization of human placental macrophages has been performed only in isolated cells. Therefore, circulation of macrophages within 3D placenta-on-a-chip models may help us understand the role of immune cells in human placental development and pathophysiology.

A better understanding of the formation and function of the placenta will be gained by exposing in vitro models to oxygen levels encountered by the tissue in vivo. During pregnancy, placental oxygen concentrations change dramatically, with earlier timepoints (up to 10 weeks) associated with low oxygen (1%–2% O_2_, 20 mmHg) and increasingly higher oxygen levels in the later stages (8% O_2_, 60 mmHg in the second trimester) [79]. Physiological oxygen concentrations relevant to placental tissue have been considered in only one in vitro study [27] to date, and to the best of our knowledge. Therefore, additional experimental interrogations are needed to elucidate in vivo placenta adaptation to a hypoxic environment.

The incorporation of biological factors present in the bloodstream of patients with pregnancy complications can enhance our knowledge on placental dysfunction. Together with the use of patient-derived primary cells, maternal blood constitutes a valuable resource of factors (principally proteins and hormones) that can influence placental function. For instance, the blood composition of patients with pre-eclampsia differs from that of individuals with uncomplicated pregnancies in the levels of sex steroids [80], pro- and anti-angiogenic factors [81], and inflammatory modulators [82]. Thus, perfusion of vascularized microfluidic placental models with blood serum obtained from women with adverse pregnancies could help to understand the etiology of placental dysfunction and to identify new therapeutic targets for future clinical applications.

Although microfluidic technologies reduce the required materials and costs compared with other culture systems, the use of small volumes could represent a limitation in the detection sensitivity of analytes. To overcome this, the integration of biosensor technologies can enhance sensing capabilities and analysis of real-time responses in on-a-chip placental models, facilitating studies on drug response and placental development [83]. Biosensors are analytic devices that comprise a biological sensing element connected to a transducer capable of providing a measurable signal. Based on the type of transducer, biosensors can be classified as optical, electrochemical, or mechanical sensors. Electrochemical-based TEER measurement, usually employed in transwell systems [26], is now becoming relevant to assess placental barrier integrity on-a-chip. For example, Schuller et al. generated a placenta-on-a-chip system containing a porous PET membrane with interdigitated electrodes that allowed for real-time and noninvasive analysis of impedance across a trophoblast monolayer, revealing the permeability of the tissue barrier during exposure to various types of nanoparticles [84].

## 7. Conclusions and Future Perspective

Placenta-on-a-chip models have recently contributed enormous progress to our understanding of human placental biology. These models overcome the ethical limitations linked to pregnancy-related studies and can be used to elucidate the pathophysiology of placental disease and act as tools for examining maternal-fetal barrier function and exchange. Biomimetic in vitro systems minimize the need for animal studies, bridge the gap between human and animal physiology, and comply with the 3Rs of research. Recent advances in in vitro approaches have enabled the arrangement of human placenta components with control over geometry and fluidics; however, many challenges are yet to be resolved. Certain aspects of placental physiology (cell types, physical features, mechanical cues) are yet to be considered in the future for the creation of physiologically relevant models. Specifically, recapitulation of the physiological features of the native tissue will help to address several open questions and advance our understanding of the mechanisms underpinning normal and pathological placental development; transport of drugs, immune cells, and hormones across the barrier and the endothelium of fetal vasculature; and response to environmental factors and pathogenic infections.

## Figures and Tables

**Figure 1 micromachines-12-00884-f001:**
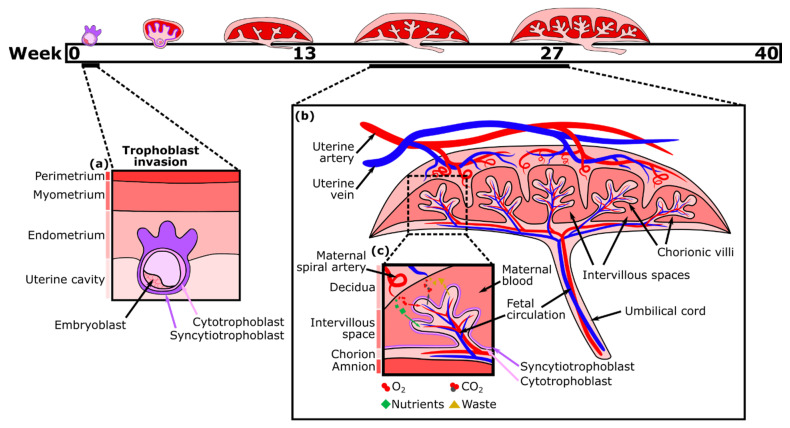
Placental development timeline, trophoblast invasion, and mature placental structure. (**a**) Diagram of trophoblast invasion around day 9, wherein the syncytiotrophoblast layer surrounding the embryoblast begins to invade the endometrium. (**b**) Mature placental structure showing maternal and fetal vasculature. (**c**) Focus on exchange of nutrients between open maternal blood and closed fetal circulation across the two trophoblastic layers.

**Figure 2 micromachines-12-00884-f002:**
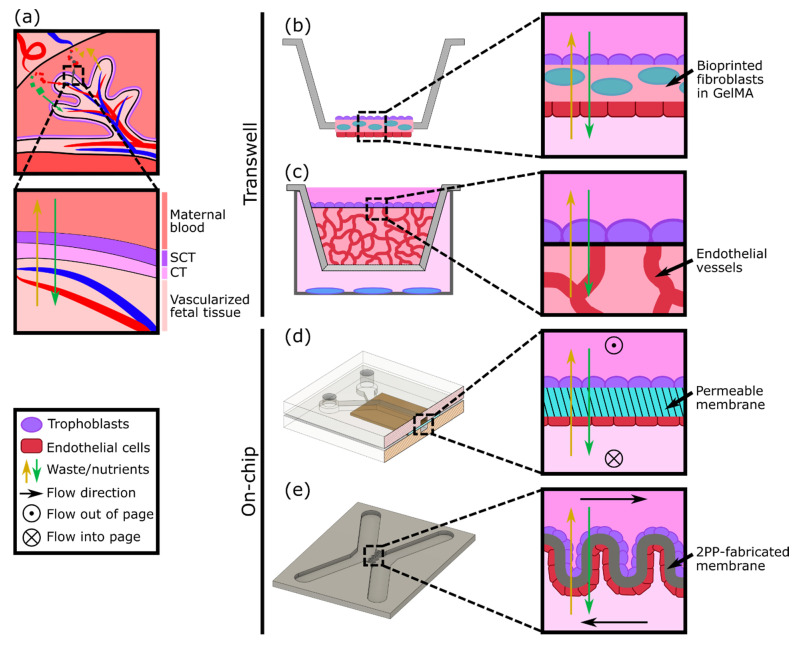
Modeling the placental barrier in vitro. (**a**) Diagram of transport between maternal and fetal blood supplies across the syncytiotrophoblast (SCT) and cytotrophoblast (CT). (**b**) A modified transwell model using a bioprinted layer of fibroblasts with trophoblasts and endothelial cells cultured on either side (based on design in [26]). (**c**) A transwell model with a layer of trophoblasts cultured on top of vasculature in a 3D gel matrix. Other cell types (blue) can be cultured below the transwell to test the effect of cell secretions (based on design in [27]). (**d**) Endothelial cells and trophoblasts can be cultured on either side of a permeable membrane in a PDMS microfluidic device with flow (based on designs in [17,28]). (**e**) Endothelial cells and trophoblasts may also be cultured on either side of a 2PP-fabricated membrane to achieve different geometries (based on design in [29]).

**Figure 3 micromachines-12-00884-f003:**
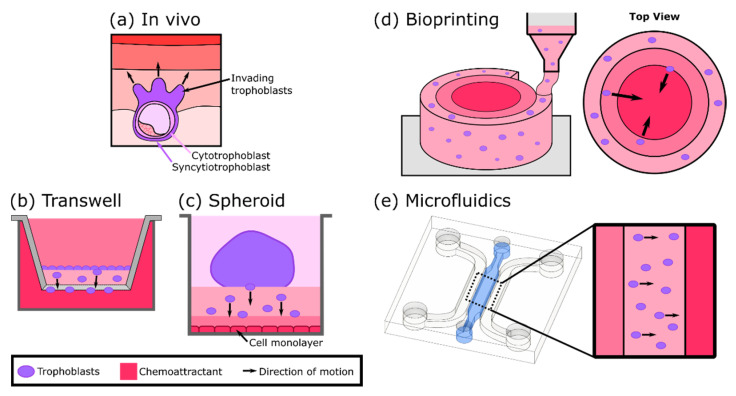
Summary of trophoblast invasion models. (**a**) Diagram of trophoblast invasion of the endometrium in vivo. (**b**) A transwell model with a monolayer of trophoblasts invading a gel towards the chemoattractant in the well (based on designs in [32,33,34,35]). (**c**) A spheroid model in a well plate with trophoblasts invading the underlying gel towards a monolayer of cells (based on design in [36]). (**d**) A bioprinted model consisting of concentric rings of a gel, with trophoblasts in the outermost layer and a chemoattractant in the innermost layer (based on designs in [37,38,39]). (**e**) A microfluidic device with trophoblasts suspended in a gel in the center channel with medium flowing through the channels on either side (based on design in [40,41]). The chemoattractant is included in only one media channel in this design.

**Figure 4 micromachines-12-00884-f004:**
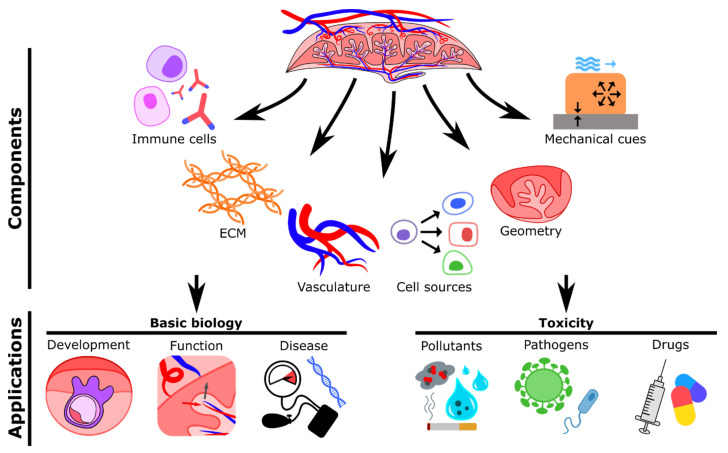
Summary of important components and possible applications of in vitro placental models. Immune cells may include Hofbauer cells or macrophages. Extracellular matrix (ECM) may be composed of collagen, laminin, fibronectin, or glycoproteins. Vascularization may be achieved via patterned microvessels, sacrificial molds, or self-assembled networks. Cell sources include cell lines, primary cells, or stem cells. Different geometries may be created by using 2D artificial membranes or two-photon polymerization (2PP). Relevant mechanical cues include fluid flow, matrix stiffness, and substrate stiffness. Possible applications in basic biology include studying the mechanisms of normal and pathological placental development and function. In vitro models may also be applied to study the transport and effects of pollutants, pathogens, and drugs.

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
