# Peer review of "Modelling the Human Placental Interface In Vitro—A Review"

_micromachines, 2021, doi:10.3390/mi12080884_

Round 1
Reviewer 1 Report
This minireview by Cherubini et al. covers the development of placenta during pregnancy, current in vitro models to study placenta and their applications, and the key components of an ideal on-chip human placenta model. The paper is well-constructed and presented in general. Here are some suggestions for the authors to improve the manuscript.
- Please also prepare a figure to help explaining different types of in vitro models of trophoblast invasion. The presentation of Figure 2 is very clear, which can be used as a template to prepare the new figure.
- In the paragraph starting from line 337, the authors discussed the effects of biomechanical cues on tissue development, which is great. In addition to stretching, compression and matrix stiffness, flow condition is also very important to the formation and development of the maternal-fetal interface. Some discussion on the in vivo flow condition, the impact of shear stress on tissue formation, and how the flow can be controlled in the in vitro placenta models would be of readers’ interest.
- As the sample volume collected from a microfluidic platform is limited compared to conventional dish/well culturing, another important aspect of an ideal human placenta-on-chip model is on-chip testing/sensing. The authors may include some further discussion on this topic. The authors have mentioned a study (ref 26) that used electrical impedance to measure the permeability of the placenta barrier, which is a good example of on-chip sensing. Here are some publications that can be referred to (DOI: 10.1016/j.trsl.2019.05.002, DOI: 10.1016/j.snb.2020.127946).
- Please correct the numerical order of the sections.
Author Response
Dear Editor and Reviewers,
Thank you for giving us the opportunity to submit a revised version of the manuscript. We are grateful to the Reviewers for their insightful comments. We have incorporated the changes to reflect the suggestions provided by the Reviewers. All changes are marked up in the “Track Changes” function. We hope that these changes allow acceptance of the manuscript.
Here is a point-by-point response to the Reviewers’ comments:
Reviewer 1
- Please also prepare a figure to help explaining different types of in vitro models of trophoblast invasion. The presentation of Figure 2 is very clear, which can be used as a template to prepare the new figure.
As suggested by the Reviewer, we have prepared and added a new figure (Figure 3) illustrating the different in vitro models of trophoblast invasion discussed in this mini-review.
- In the paragraph starting from line 337, the authors discussed the effects of biomechanical cues on tissue development, which is great. In addition to stretching, compression and matrix stiffness, flow condition is also very important to the formation and development of the maternal-fetal interface. Some discussion on the in vivo flow condition, the impact of shear stress on tissue formation, and how the flow can be controlled in the in vitro placenta models would be of readers’ interest.
We agree with the Reviewer and as suggested we have incorporated some sentences (from line 369) emphasizing the importance of the flow in placenta development and its integration in the in vitro models.
- As the sample volume collected from a microfluidic platform is limited compared to conventional dish/well culturing, another important aspect of an ideal human placenta-on-chip model is on-chip testing/sensing. The authors may include some further discussion on this topic. The authors have mentioned a study (ref 26) that used electrical impedance to measure the permeability of the placenta barrier, which is a good example of on-chip sensing. Here are some publications that can be referred to (DOI: 10.1016/j.trsl.2019.05.002, DOI: 10.1016/j.snb.2020.127946).
We thank you the Reviewer for this suggestion. We have added a paragraph (from line 414) discussing the integration of biosensor in placenta in vitro models.
- Please correct the numerical order of the sections.
We thank you the Reviewer for pointing this out. We have corrected the numerical order of the sections.
Reviewer 2 Report
The authors provide a review of conventional and innovative in vitro models of the human placenta.
In general the manuscript provides a good overview, it is clear and well-written. The work adds a good contribution to the field given the only other recent review found in the literature (from 2019) focuses just on drug transport studies. Here, while still reviewing in vitro models of the placenta used to study drug transport, the Authors expanded their scope to include studies of invasion, dysfunction, infection, and others. The authors also provide a good introduction to the anatomo-physiology of the placenta and to conventional in vitro platforms. Overall, a solid piece of work.
There aren't any significant changes I can recommend.
Some minor typos in the text should be corrected, example:
line 32: 'However' is a better word than 'Nevertheless' here.
lines 97-100: This sentence would benefit from re-writing. It is confusing.
line 100: 'ad'
line 115: I don't think 'proper' is the best word here.
line 163: microvilli projections
line 164: 'when compared to' would be better than 'rather than' here in my opinion.
line 173: '... findings from Blundell's model'
line 226: 'Besides bioprinting' - there's one extra comma
lines 337-339: This sentence is not very well written. I would say. "Together with appropriate cell types, an adequate structural and spatial arrangement of cells ... is needed to mimic, on-chip, the in vivo maternal-fetal interface in a more physiologically relevant way"
Author Response
Dear Editor and Reviewers,
Thank you for giving us the opportunity to submit a revised version of the manuscript. We are grateful to the Reviewers for their insightful comments. We have incorporated the changes to reflect the suggestions provided by the Reviewers. All changes are marked up in the “Track Changes” function. We hope that these changes allow acceptance of the manuscript.
Some minor typos in the text should be corrected.
line 32: 'However' is a better word than 'Nevertheless' here.
lines 97-100: This sentence would benefit from re-writing. It is confusing.
line 100: 'ad'
line 115: I don't think 'proper' is the best word here.
line 163: microvilli projections
line 164: 'when compared to' would be better than 'rather than' here in my opinion.
line 173: '... findings from Blundell's model'
line 226: 'Besides bioprinting' - there's one extra comma
lines 337-339: This sentence is not very well written. I would say. "Together with appropriate cell types, an adequate structural and spatial arrangement of cells ... is needed to mimic, on-chip, the in vivo maternal-fetal interface in a more physiologically relevant way"
We thank you the Reviewer for pointing these out. All spelling and grammatical typos, and sentence rewriting suggested by the Reviewer have been corrected or modified.
Reviewer 3 Report
The presented mini-review is devoted to the actual problem of modeling the human placental interface in vitro. The article is written in good English, good and understandable language, contains an explanation of all terms for readers working in related scientific fields. The article provides good information that explains the features of the placental structure formation process, and also substantiates the relevance of the posed problem of modeling the placental interface in vitro.
In my opinion, the article can be accepted for publication, but I have a few recommendations that I can ask as an engineer-physicist working in the development of bioprinting technologies:
- In section “Engineering an Ideal Human Placenta-On-Chip ”, it seems to me appropriate to give the authors' opinion on what modern technologies (including varieties of three-dimensional printing options) can be used to form such devices. Perhaps (at the discretion of the authors) it makes sense to provide a small table with a variety of methods and technologies, and indicate for which part of the Placenta-On-Chip a particular method or technology can be applied. This will allow the reader to assess their capabilities, as well as once again familiarize themselves with modern methods and technologies for the formation of such systems.;
Other corrections:
- Section “5. Engineering an Ideal Human Placenta-On-Chip”must be numbered 6;
- Section “5.Conclusions and Future Perspective" must be numbered 7;
Author Response
Dear Editors and Reviewers,
Thank you for giving us the opportunity to submit a revised version of the manuscript. We are grateful to the Reviewers for their insightful comments. We have incorporated the changes to reflect the suggestions provided by the Reviewers. All changes are marked up in the “Track Changes” function. We hope that these changes allow acceptance of the manuscript.
- In section “Engineering an Ideal Human Placenta-On-Chip ”, it seems to me appropriate to give the authors' opinion on what modern technologies (including varieties of three-dimensional printing options) can be used to form such devices. Perhaps (at the discretion of the authors) it makes sense to provide a small table with a variety of methods and technologies, and indicate for which part of the Placenta-On-Chip a particular method or technology can be applied. This will allow the reader to assess their capabilities, as well as once again familiarize themselves with modern methods and technologies for the formation of such systems.;
First of all, we thank the Reviewer for the positive comments. In regards to discussing bioprinting technologies, we hope that given the addition of the new Figure 3 this will clarify for the Readers the type of technology discussed in the article. This Review is not intended to give an in-depth overview of either microfluidics nor bioprinting technologies per se, but rather focuses on the different in vitro models generated of the placenta.
Other corrections:
- Section “5. Engineering an Ideal Human Placenta-On-Chip”must be numbered 6;
- Section “5.Conclusions and Future Perspective" must be numbered 7;
We have corrected the sections accordingly. Thank you for pointing out this error.
Round 2
Reviewer 1 Report
The authors have addressed my questions. I have no more comments.